# Inter-individual differences in human brain structure and morphology link to variation in demographics and behavior

**Alberto Llera[1,2,3]\*, Thomas Wolfers[1,2,3], Peter Mulders[1,2,3,4], Christian F Beckmann[1,2,3,4,5]**

[1]Radboud University Nijmegen, Nijmegen, Netherlands; [2]Centre for Cognitive Neuroimaging, Donders Institute for Brain, Cognition and Behavior, Nijmegen, Netherlands; [3]Department of Cognitive Neuroscience, Radboud University Medical Centre, Nijmegen, Netherlands; [4]Department of Psychiatry, Radboud University Medical Center, Nijmegen, Netherlands; [5]Oxford Centre for Functional Magnetic Resonance Imaging of the Brain (FMRIB), University of Oxford, Oxford, United Kingdom

**Abstract** We perform a comprehensive integrative analysis of multiple structural MR-based brain features and find for the first-time strong evidence relating inter-individual brain *structural* variations to a *wide range* of demographic and behavioral variates across a large cohort of young healthy human volunteers. Our analyses reveal that a robust 'positive-negative' spectrum of behavioral and demographic variates, recently associated to covariation in brain function, can already be identified using only structural features, highlighting the importance of careful integration of structural features in any analysis of inter-individual differences in functional connectivity and downstream associations with behavioral/demographic variates.
DOI: https://doi.org/10.7554/eLife.44443.001

**\*For correspondence:**
a.llera@donders.ru.nl

**Competing interests:** The authors declare that no competing interests exist.

## Introduction

Understanding individual human behavior has attracted the attention of scientists and philosophers since antiquity. The first quantitative approach intended to deepen such understanding dates to the first half of the 19-th century when skull measures were related to human behavior or cognitive abilities (*Simpson, 2005*; *Fodor, 1983*). Technical, intellectual and clinical advances in the last two centuries allow us to now accurately quantify brain structure and function (*Lerch et al., 2017*; *Huettel et al., 2004*; *Friston et al., 2002*; *Woolrich et al., 2004*; *Rorden et al., 2007*), and to summarize certain 'aspects' of human behavior by means of standardized tests. Such advances facilitate exploratory statistical learning analyses to uncover previously hidden relationships between brain features and human behavior, demographics or pathologies (*Poldrack and Farah, 2015*). These developments are expected to be pushed even further with the emergence of the big data magnetic resonance imaging (MRI) epidemiology phenomenon (*Van Essen et al., 2013*; *Collins, 2012*), and some examples of such expectations have already reported associations with blood-oxygen-level dependent (BOLD) brain function (*Finn et al., 2015*; *Smith et al., 2015*); for example, functional connectivity patterns can be used to identify individuals (*Finn et al., 2015*), predict fluid intelligence (*Finn et al., 2015*), or describe a mode of functional connectivity variation that relates to lifestyle, happiness and well-being (*Smith et al., 2015*).

Although the brain's structural-functional relationships are not yet fully understood, linking structure to behavior is essential for either type of imaging modality to be fully interpretable as an imaging phenotype. Furthermore, given the long-term character of some demographic variables (e.g.

**eLife digest** For years, scientists have tried to explain human behavior by measuring brain characteristics. During the first half of the 19th century, craniometry, the science of taking measurements of the skull, was a popular field of research and cognitive abilities as well as many behaviors were associated with different skull sizes and shapes. Although craniometry has been broadly discredited as a science, the study of brain structure and function, and their correlation to human behavior, continues to this day.

Currently, one of the most powerful tools used in the study of the brain is magnetic resonance imaging (MRI), which relies on strong magnetic fields and radio waves to produce detailed imaging. These images can provide functional information, by measuring changes in blood flow to different parts of the brain, as well as structural information such as the amount of gray or white matter or the size of different brain regions. Many studies have shown correlations between functional MRI (fMRI) data and behavioral and demographic traits, such as years of education, lifestyle habits or stress. Another advance in the study of the relationship between behaviors and the brain has been the emergence of better statistical analysis tools thanks to increasing computing power. These tools have made it possible to integrate data from different sources and analyze many variables at the same time, allowing patterns to emerge that would have been previously missed.

Llera et al. have analyzed a large dataset from young healthy volunteers to show that changes in behavioral traits can be predicted by brain structure, and not just by brain function as previously shown. Different types of brain structural data, including what the surface of the brain looks like and relative volumes of gray and white matter, were integrated and analyzed, and correlations between changes in these variables and changes in the demographic and behavioral traits of the subjects were found. Previously, a robust relationship had been established between specific patterns of connections and activity in the brain and a group of characteristics such as life satisfaction, working memory, weight and strength, loneliness, family history of drugs and alcohol use, etc. Llera et al. show that this relationship also holds between the traits and structural brain data. As an example, there is a positive correlation between changes in the number of years of education and the income of the subjects and changes in a pattern of integrated structural data that include the amount of gray matter, white matter integrity and size of specific brain structures. Given these findings it becomes important to reconsider whether differences between individuals previously attributed to brain function could simply explained by the shape or size of the brain and its parts.

These findings show that physical brain characteristics, including its size or the shape of its surface, could predict information such as individuals' lifestyle decisions or their income; also implying that these characteristics are not simply a product of brain function. The results also demonstrate the power of combining different types of brain data to predict patterns in behavior.
DOI: https://doi.org/10.7554/eLife.44443.002

overall happiness), we hypothesize that different brain structural features, such as regional variation in the density of gray matter or subject-dependent degree of cortical expansion, should also reflect these relationships. To test these hypotheses, in this work we make use of the large quantity of high quality behavioral and neuroimaging data collected by one of the big data initiatives, the Human Connectome Project (*Van Essen et al., 2013*) (HCP). The HCP sample includes detailed structural imaging, diffusion MRI, resting-state and several different functional MRI tasks for each subject. Furthermore, the availability of more than 300 behavioral and demographic measures (*Van Essen et al., 2012*) allows the post-hoc exploration of a wide range of associations (*Groves et al., 2011*). We further hypothesize that behavioral variations can be explained by more general brain structure variations than isolated single feature variations (e.g. cortical thickness variations); we consequently extract multiple structural features from the different MR modalities and perform a simultaneous analysis by linked independent component analysis (Linked ICA; *Groves et al., 2011*; *Groves et al., 2012*). Linked ICA is a Bayesian extension of Independent Component Analyses developed for multi-modal data integration, where multiple ICA factorizations are simultaneously performed and all of them share the same unique mixing matrix. Such analyses increase statistical power by evidence integration across different features (*Wolfers et al., 2017*; *Doan et al., 2017*) and have been

shown to be powerful in identifying correlated patterns of structural and diffusion spatial variation that can then be studied in relation to individual behavioral and demographic measures (*Doan et al., 2017*; *Douaud et al., 2014*; *Francx et al., 2016*). Although similar analyses have been previously performed (*Douaud et al., 2014*), in this work we benefit from the unique characteristics of the data sample; we consider brain and behavioral data from close to 500 'healthy young adults' which reduces common pathology- and age-related variance and increases the power to detect associations due to normal cross-sectional variability.

Our results support the hypothesis that structural brain features are strongly associated with demographic and behavioral variates. Interestingly, the most relevant mode of inter-individual variations across brain structural measures identified through the multi-modal data fusion approach maps on to recent findings obtained using functional MRI data from the same HCP cohort. In particular, our findings closely resemble the 'positive-negative' set of behavioral measures identified in *Smith et al. (2015)* on the basis of functional (co-)variations. Using post-hoc analysis of the functional and structural modes we show that inter-individual differences attributed to brain function need to be reconsidered taking into account variations in brain structure across the cohort.

## Results

The multi-modal structural brain data analyses (*Figure 1*, operations A and B) resulted in a total of 100 collections of component maps, each of which can be represented by a collection of 7 spatial maps covering the gray-matter space (voxel-based morphometry feature (VBM)), diffusion skeleton space (Fractional Anisotropy (FA), Mean Diffusivity (MD) and Anisotropy Mode (MO) features), cortical vertex space (cortical thickness (CT) and pial area (PA) features) and a voxel-wise map of the Jacobian deformation (JD). In addition, each collection of maps is associated with a single vector of contributions that describe the degree to which a given collection is 'driven' by the different modalities (feature loadings). Finally, each collection is associated with a single vector that describes how each individual subject contributes to the component (subject loadings). Post-hoc linear correlation analyses of these subject contributions with behavioral measures identified, after FDR correction (*Figure 1*, operations C, D and E, FDR corrected $q < 2.2 \times 10^{-4}$), a total of 155 significant brain-behavior correlations, summarized by 30 components reflecting at least one significant relationship to behavior. We provide the full results in *Supplementary file 2* and a brief summary in the bottom left panel of *Figure 1* where we color code the significant Pearson correlation values for the components showing at least one Bonferroni corrected (Bonferroni corrected $q < 1.4 \times 10^{-6}$) significant correlation to a behavioral or demographic measure.

Only a single component (number 6) shows strong associations across a broad set of behavioral domains (48 measures) and across all structural modalities (i.e. is not dominated by one of the structural data types in that no single modality contributes >50% to the total variance of the component). The relative contributions from the different modalities are 22% for VBM, 6% JD, 15% FA, 23% MD, 20% MO, 7% CT and 4% PA (*Appendix 1—figure 1*), reflecting a dominance of gray matter densities and diffusion measures and a lower involvement of the purely morphometric and cortical measures. Relating the behavioral associations, in *Figure 1* bottom right panel we provide a summary of the behavioral measures significantly correlating with component six as well as the corresponding Pearson correlation values. Note that in the cases where several measures are grouped together we report their mean correlation value - full results are given in *Supplementary file 2*. We observe that component number six relates to various behavioral scores including working memory, language function and general wellbeing (life satisfaction, social support). In *Figure 2* we present the associated spatial maps: VBM measures are most heavily weighted in bilateral orbitofrontal cortex, temporal pole, lingual gyrus and the putamen (first row). Morphometric differences (JD features) load into temporal lobes, caudate and brainstem (second row), and white matter tracts do most heavily weigh onto the internal capsule, anterior thalamic radiation and the anterior corona radiata (3rd, 4th, and 5th rows). Cortical effects (6th and 7th rows) are largely associations with multi-modal association cortex that show effects whereas primary sensory cortices are not implicated. Note that the involvement of areas such as the putamen and lingual gyrus are relevant to explain the behavioral relationships found with working memory and word processing. Furthermore, the involvement of structural connections between subcortical and prefrontal areas as well as the orbitofrontal cortex and temporal poles could explain the link to more complex functions such as emotional support or life satisfaction.

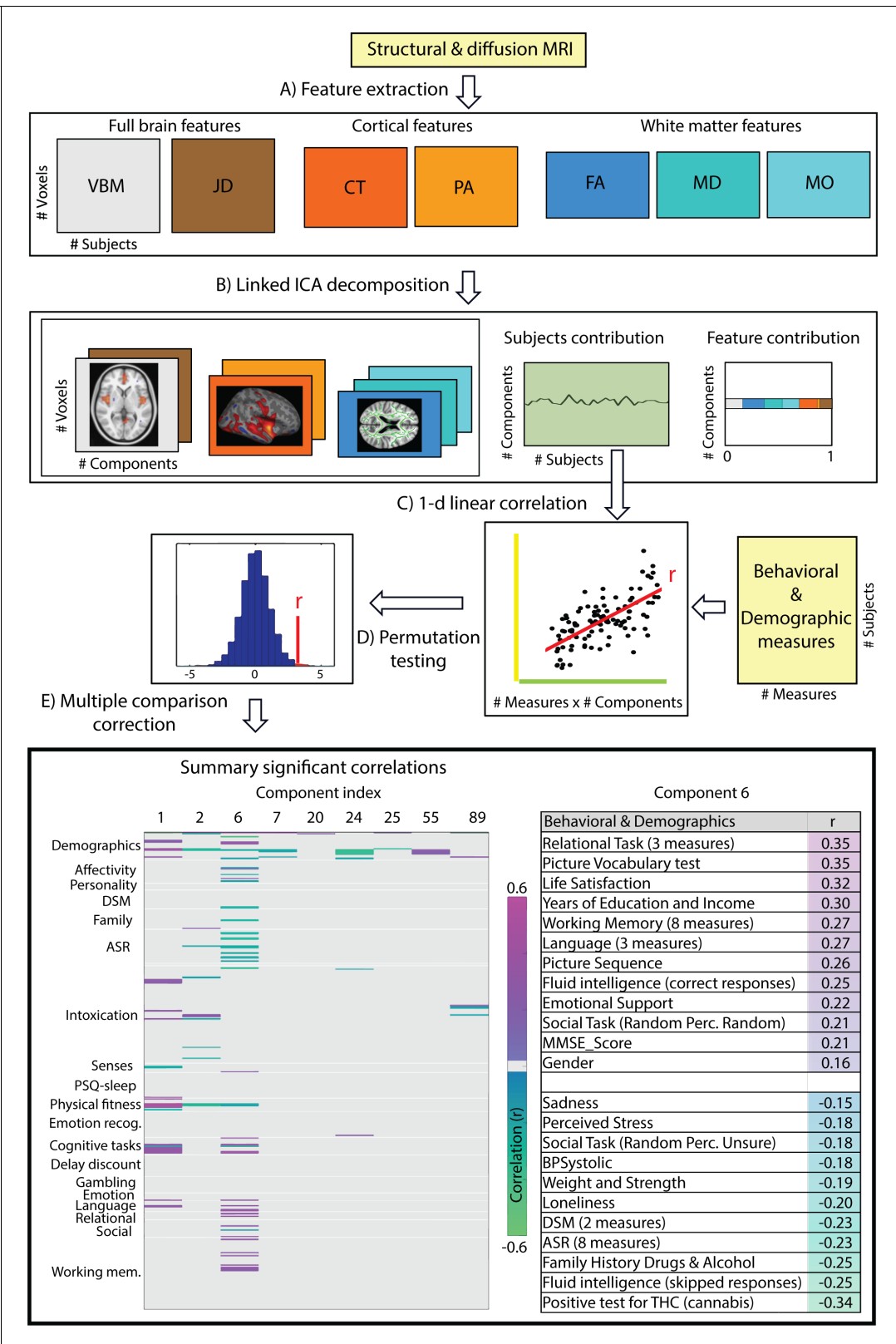

**Figure 1.** Data processing pipeline and main results. (**A**) Structural and diffusion-weighted MRI data are used to extract relevant features, that is, Voxel-Based Morphometry (VBM), Fractional Anisotropy (FA), Mean Diffusivity (MD), Anisotropy Mode (MO), Cortical Thickness (CT), Pial Area (PA) and Jacobian Determinants (JD). (**B**) These features are used as input to the Linked ICA algorithm. (**C**) Subject loadings of each independent component are fed together with the behavioral/demographic measures into a correlation analysis. The bottom left panel presents demographic and behavioral

*Figure 1 continued on next page*

*Figure 1 continued*

measures grouped by categories (y-axis), and a representative set of components reflecting significant correlation with at least one behavioral measure (x-axis). The color-scale encodes the Pearson correlation coefficient and only significant correlations are color-coded. In the bottom right panel, we present a summary of component number six significant correlations to behavioral and demographic variates where the behavioral measures are grouped and ordered according to a decreasing correlation value. These results resemble a mode of structural variation that links to and extends the 'positive-negative' behavioral spectrum previously attributed to functional connectivity variations (*Smith et al., 2015*).

DOI: https://doi.org/10.7554/eLife.44443.003

Note that each component considers structural multi-modal characterizations of the brain where each modality contains unique information and together builds into a multi-modal multivariate component. Consequently, although these results are hard to interpret as being nested into the same space as (functional) canonical brain networks, the structural weighting in gray matter modalities in orbitofrontal and temporal cortex, in conjunction with the white matter tracts that connect those regions, is a clear indication of an underlying network structure relating to component six and consequently, to its behavioral associations.

Several other components also reflect behavioral patterns worth recognizing. In *Figure 3* we report spatial maps associated with the components showing at least one Bonferroni corrected significant relationship with any behavioral measure ($p<1.4\times10^{-6}$). For components 1, 2 and 89 we show spatial maps for all modalities contributing (*Appendix 1—figure 1*). In order to provide a clearer interpretation, for the other components we decided to show a selection of the relevant modalities and full NIfTI maps are separately available as supplementary material. Component one relates mainly to gender, physical strength and language and it is defined by significant changes in gray matter density (VBM measures) and cortical areal expansion (PA measure). Its associated spatial patterns appear to in fact reflect brain size and cortical area differences in both temporal lobes. Similarly, component two is driven by VBM maps and correlates with variations in gender, age, height, weight and strength. Its spatial extent includes the paracingulate gyrus and bilateral insular and opercular cortex. Components 7, 24, 25, 29 and 55 are driven by at least three feature modalities and they map into gender, weight, body mass and height. Component seven maps into gender and shows cingulate gyrus and insular cortex.

Component 24 maps into weight and body mass and is mapped into putamen, intracalcarine cortex and thalamus. Components 25 and 29 relate height with the inferior temporal gyrus and the cerebellum together with strong DWI weightings in the brainstem. Component 55 relates to weight and maps into the precentral gyrus and asymmetric differences in DWI measures. Number 89 maps VBM and JD into hematocrit and involves the lingual and occipital fusiform gyrus. Finally, component 20 maps JD and VBM in the posterior midline into age and relationship status.

Although another set of components show associations to behavior, these are limited to a single modality and/or a small set of behavioral variates (*Supplementary file 2*). Many of these components show simple relationships to overall size measures such as weight, body mass (BMI) or height, and the associations are weaker than those reported above; we consequently decided to not further discuss their spatial extent in this work and we provide full NIfTI images as supplementary material.

To validate the robustness of the presented results to the model order choice we performed analyses at different dimensionalities and observe that especially lower indexed components are highly reproducible. In particular, component number six is recovered at dimensionalities 90 and 110 with a subject mode correlation value of around r ~ 0.9 (details can be found in Appendix 1, section 'Robustness: model order'). Regarding the influence of purely morphometric differences in the analyses, a comparative analysis excluding the JD revealed essentially unaltered brain-behavior associations. Analysis of the JD feature in isolation showed that no fully corrected significant association to the reported positive-negative structural mode is found when considering uniquely morphometric differences, even if considering several components together. However, uncorrected statistics suggests that information of the positive-negative mode could already be present at the morphometric level. These results are presented in Appendix 1, section 'Robustness: analyses without Jacobians', and 'Robustness: Analyzing morphometric differences'.

Given the similar associations to behavior found between the presented structural mode (component 6) and the 'positive-negative' functional mode reported in *Smith et al. (2015)*, and since both results are obtained using the HCP sample, we quantified the linear relation between them. With the

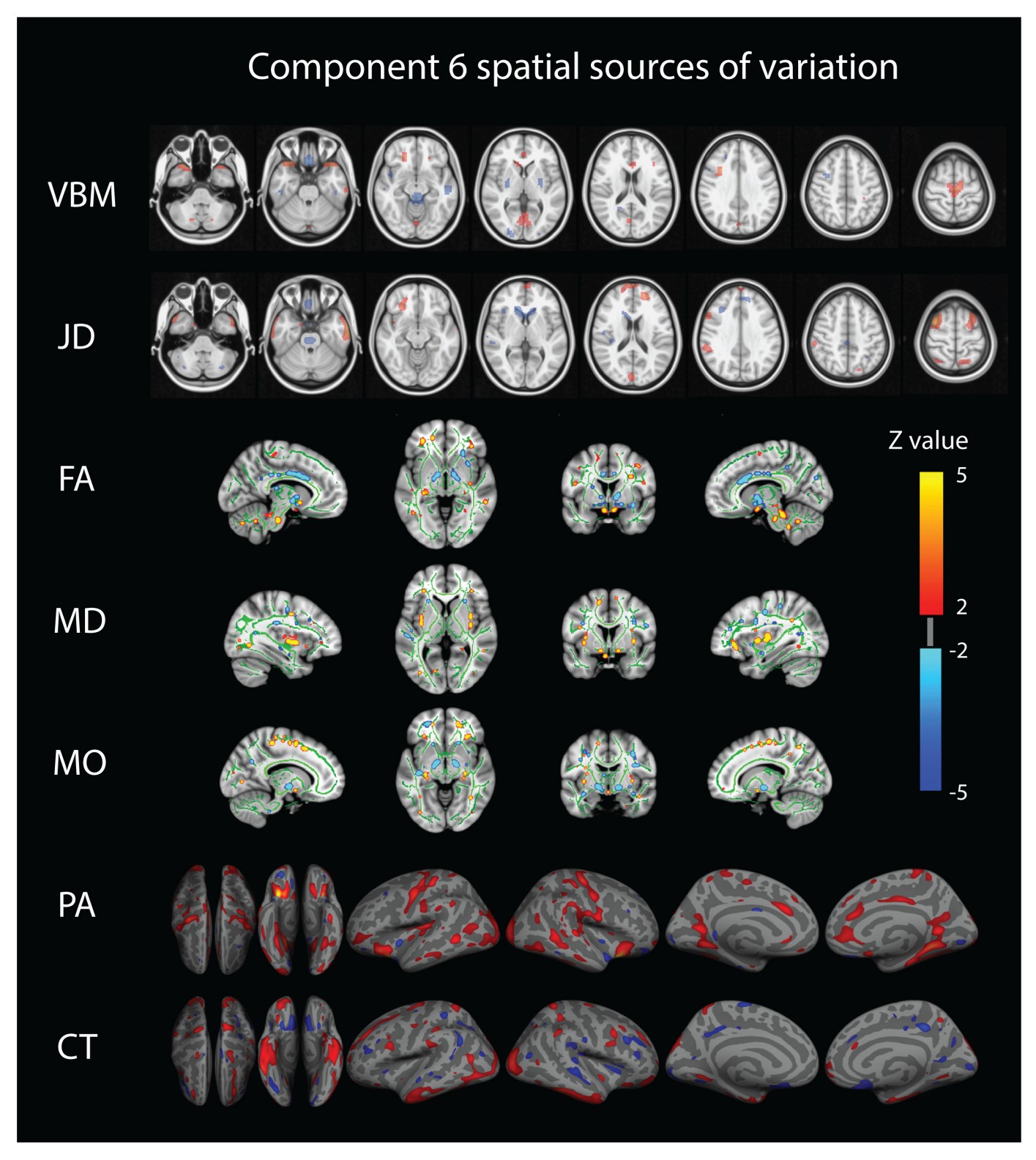

**Figure 2.** Component number six feature sources of variation. From top to bottom we visualize the VBM (Voxel Based Morphometry), JD (Jacobian Determinants), FA (Fractional Anisotropy), MD (Mean Diffusivity), MO (Mode of Anisotropy), PA (Pial Area), and CT (Cortical Thickness) spatial maps. For improved visualization, each modality has been thresholded at a z-value of 2. This mode of structural variation, component 6, that strongly reflects a

*Figure 2 continued on next page*

*Figure 2 continued*

'positive-negative' behavioral spectrum, links to a wide range of brain regions across structural modalities and might reflect the structural multi-modal foundation of a functional brain network linked to these variations that has been earlier identified.

DOI: https://doi.org/10.7554/eLife.44443.004

analysis restricted to the 421 subjects common to both studies, we found that the structural and the functional subject modes are significantly correlated (r = 0.4643, $r^2$ = 0.21, permutation p<$10^{-5}$). Post-hoc correlation analyses to behavior replicated the original functional positive-negative mode by identifying 60 functional-behavioral relationships (*Smith et al., 2015*); we found that 22 of these behavioral measures are also associated to the structural mode and that there is no significant difference in the correlation values provided by the functional or the structural analyses at these intersecting behavioral measures (details can be found in Appendix 1, section: 'On the power of structural and functional associations to behavior').

To identify the linear dependence between the behavioral/demographic modes obtained from functional and structural data we used a generalized linear model (GLM). We regressed the structural mode from the functional one and performed post-hoc linear correlation analysis of the residualised functional mode relative to behavioral variates as in *Smith et al. (2015)*. Note that structural features – due to the necessary co-alignment within the functional pipelines – acts as a mediator and therefore could induce significant imaging-to-behavior associations (also see *Bijsterbosch et al., 2018*). Conversely, however, the structural features enter into the cross-subject analysis without any possible cross-talk from functional data, so that there is no possible interference from functional to structural features. The post-hoc correlation analysis of the residualised functional mode to behavior revealed a significant decrease in correlation (mean r decrease = 0.078, p<0.01) that result in the structural mode removing 73% of the 60 associations originally found using functional data. The remaining 16 significant relationships involve measures as handedness, education, tobacco use, list sorting, delay discount, and intelligence. As such, the two modes are significantly overlapping.

We also performed a structural-functional Linked-ICA analyses where we added partial correlation matrices obtained from resting state fMRI to the set of originally considered structural features. We selected functional fMRI features to match *Smith et al. (2015)*; details on data availability and processing are provided in Appendix 1, section 'Individual features pre-processing'. This structural-functional analysis recovered the positive-negative mode reported in the originally reported multi-modal structural analyses. More concretely, we found a component, number 16, significantly correlating (r = 0.89, p<$10^{-5}$) with the mainly reported structural mode (component 6). The contribution of each modality to this mode equals 20% for VBM, 15.6% for FA, 24.4% for MD, 23.9% for MO, 7% for CT, 3% for PA, 5% for JD and 0.0012% for the functional partial correlation feature. While all structural features reflect approximately the same contribution as in the original structural analyses, it is interesting that the functional data does marginally contribute to the found mode, suggesting that structure in its own can explain the positive-negative behavioral mode.

As a final step in our analysis we performed a causal analysis between the structural mode (component 6) and the functional mode reported in *Smith et al. (2015)*, on the basis of calculating pairwise likelihood ratios (*Hyvarinen and Smith, 2013*) between the function-to-structure and structure-to-function model. This analysis estimated a likelihood ratio of ~0.04, that is a significant structure to function causation effect (p<0.0025, using permutation testing) (*Hyvarinen and Smith, 2013*) (for details see Appendix 1, section 'Testing structure-function causal effects'). Replication of these causal results was achieved by performing an analogous multi-modal structural Linked ICA analyses considering this time the HCP1200 sample, and including only the subjects not used in the originally considered HCP500 sample. The analyses revealed a component that, restricted to the 331 intersecting subjects, showed once more a significant correlation with the positive-negative mode reported in Smith et al. (r = 0.1773, p<0.002). Furthermore, consecutive causal analyses from the structural mode to the functional mode estimated a likelihood ratio of ~0.09, that is a significant structure-to-function causation effect (p<0.005).

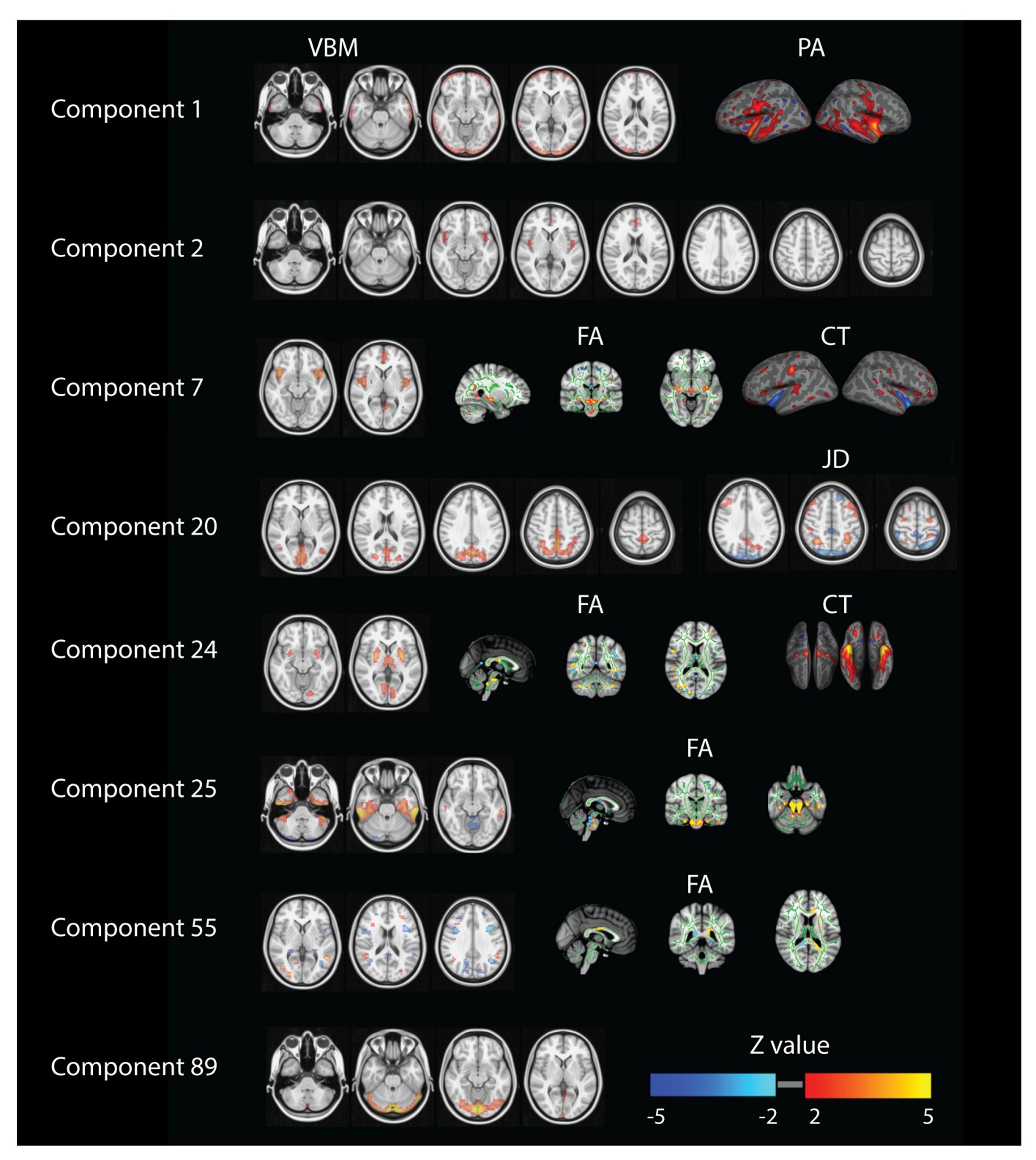

**Figure 3.** Summary of relevant modalities spatial maps associated with the components indexed in the most left column. For component one we show spatial maps for VBM and PA, and for components number 2 and 89 just VBM. For numbers 7 and 24 we present VBM, FA and CT. For number 20 we show VBM and JD and finally for numbers 25, 29 and 55 we present VBM and FA.

DOI: https://doi.org/10.7554/eLife.44443.005

## Discussion

We present a simultaneous analysis of brain structural measures that reveals how several types of behavior and demographics link to variations in such measures of brain structure. Several components detect simple associations between brain size (encoded in gray matter density and cortical area) being related to gender, strength, endurance or language function. More interestingly, we encounter a single pattern of gray and white matter covariation that is strongly associated with several measures relating to cognitive function including working memory and language function, while also being strongly related to several measures of wellbeing including life satisfaction or emotional support. Accordingly, the spatial organization of the component that relates to these measures predominantly includes regions and connections that are relevant to working memory and word processing such as the putamen and lingual gyrus (*Mechelli et al., 2000*; *Arsalidou et al., 2013*). Additionally, the inclusion of regions such as the orbitofrontal cortex and temporal poles, as well as structural connections from subcortical to prefrontal regions, could explain the link to more complex functions such as emotional support and life satisfaction. Furthermore, the mode of structural variation we report here relates to several recently reported results obtained using functional MRI. In particular, our results relate to the ones presented in *Finn et al. (2015)* since it identifies fluid intelligence measures and it also shares many behavioral measures also identified by the 'positive-negative' mode reported in *Smith et al. (2015)*. Clearly, the functional analyses presented in *Smith et al. (2015)* and the one we present here, while using entirely different MRI measurements, are both able to get at the core of the same behavioral spectrum; in fact, the structural mode and the functional mode are strongly correlated subject measures (r = 0.46). Our analyses reliably augment the spectrum of behavioral variables reported by the functional analyses by extending it with many working memory, language, relational task, ASR and DSM measures (*Figure 1* bottom right and *Supplementary file 2*). It is to note here that while the statistics reported in *Smith et al. (2015)* were obtained from a Canonical Correlation Analyses (CCA) between partial correlation matrices and all behavioral measures at once, the statistics we present here involve simple linear correlations. While the former type of analysis can benefit from the multi-variate type of analysis through the application of CCA, ensuing results can be hard to interpret. The straight-forward individual linear correlation analysis against the behavioral/demographic measures separately instead affords simple interpretation.

These findings directly look into the relationship between brain structure and function. In fact, the functional mode of variation is strongly associated with connectivity in brain areas approximately resembling the Default Mode Network (*Smith et al., 2015*) and, given the spatial extent and the strong weight of the DWI data in the structural mode we report, it seems reasonable to assume that these white matter structure variations could contribute to the functional connectivity changes reported in *Smith et al. (2015)*. Further, we found no clear spatial overlap between the reported structural mode and the cortical functional extent of the 'positive-negative' mode, suggesting that integrated functional-structural analyses should increase the sensitivity of both functional and structural analyses. Further, these results might question whether group functional connectivity measures using fMRI provide direct measures of brain connectivity or are biased due to individual structural differences that may become 'visible' in the analysis of functional cross-subject. An analogous multi-modal analysis excluding the JD feature provided equivalent results to those presented here (*Supplementary file 3*) and unimodal analysis of only the JD features (using simple ICA-based decomposition (*Beckmann and Smith, 2004*) of the single JD modality) did not provide significant correlation to the behavioral mode at the level of fully corrected statistics. These extra analyses confirm that the structural features relating to the behavioral mode are not uniquely driven by morphometric differences. The post-hoc correlation analysis of the residualised functional mode to behavior revealed a significant decrease in correlation (mean r decrease = 0.078, p<0.01) that result in the structural mode removing 73% of the 60 associations originally found using functional data. The remaining 16 significant relationships involve measures as handedness, education, tobacco use, list sorting, delay discount, or intelligence. As such, our results confirm that many associations previously attributed to functional connectivity are already present at the structural level. This could be interpreted in terms of a specialization of functional imaging towards a specific subset of behavioral measures for which it provides strong effects even after linear accounting for the structural findings, implying that not all previously identified associations can be explained through inter-individual

differences in brain structure. As such, these two modes are significantly overlapping measures that are not fully reflected in the Jacobian deformation field. The presence of residual functional associations to behavior suggests that these associations - although possibly influenced by structural variation - cannot uniquely be attributed to simple morphometric differences. These results align with recent findings by *Bijsterbosch et al. (2018)* who show that individual spatial configurations extracted from functional MRI rather than the connectivity profiles between areas seem to stronger relate to the positive-negative mode.

While the presence of residual associations could be interpreted as evidence for functional-structural integration, care needs to be taken with regards to the interpretation of these associations and changes thereof. First, note that all of these methods interrogate the linear relationships between variates. It is entire possible that the association between imaging phenotypes and behavioral/demographic measures involve non-linear relationships that remain at best incompletely accounted for within these analytical frameworks. Second, the implicit symmetry of linear correlations implies that a corresponding residualised analysis (where we regress functional variations from the structural mode) similarly removes significant associations. Indeed, in such a case only 7 out of 48 associations remain significant (relating to weight, antisocial behavior (DSM), family structure problems, relational task or adult self-report (ASR) questions, see Appendix 1, section 'On the power of structural and functional associations to behavior'). These results suggest a segregation of different structural and functional specializations towards different behavioral measures with for example, intelligence being, not only, but more related to brain function, and antisocial or relational task measures relating more strongly to brain structure.

Finally, a causal analysis revealed a significant structural to functional mode causation (*Hyvarinen and Smith, 2013*) where the likelihood of the structural mode causally influencing the functional mode (from *Smith et al., 2015*) is >20 times higher than the likelihood of the reverse causation. Although the causal model introduced in *Hyvarinen and Smith (2013)* considers the residuals after linear modeling of a pair of signals, care is advised when considering causal inference on two vectors of observations, as we cannot exclude the possibility that unobserved underlying processes simultaneously influence brain structure and function ('hidden causation'). Nevertheless, these causal findings align with the fact that cross-subject analysis of functional data typically necessitates processing of structural data (e.g. through co-registration into a common space). As such, structural variations will enter as mediating factor in any functional analysis pipeline and need to be accounted for suitably. However, there is no reverse influence of functional variations in the analysis of structural measures. Such dependencies remain poorly modeled in current analysis procedures and future work will have to focus on robustifying functional MRI analysis with regard to cross-subject variations in brain structure, for example by more advanced alignment procedures and/or through derivation of functional measures that are invariant under variations in structure. This will have important implications for the interpretation of future finding across neuroimaging 'big data' studies and will help improve our understanding of the functional-structural integration and its relation to behavioral associations.

## Materials and methods

In this work, we use data from the Human Connectome Project (HCP) N = 500 release which contains data from healthy young adults including twins and their non-twin siblings. In addition to performing more than 300 behavioral/demographic tests, each subject participated in structural, diffusion and several functional MRI recordings (*Van Essen et al., 2012*; *Elam and Van Essen, 2013*). A description of all MRI and behavioral/demographic measures included in our analysis can be found in van *Van Essen et al. (2012)* and a short description is available at https://www.human-connectome.org/storage/app/media/documentation/q3/HCP_Q3_Release_Appendix_VII.pdf; we also provide a summary of the latter in the Appendix 1, *Supplementary file 1*. Due to structure-function integration we hypothesize that different biological features such as regional variation in the density of gray matter, white matter connectivity or subject dependent degree of cortical expansion should reflect similar associations with behavior as the ones reported at *Finn et al. (2015)* and *Smith et al. (2015)*. To investigate such hypothesis, the structural MRI T1-weighted images were used to extract gray matter densities and cortical measures, using a Voxel Based Morphometry (VBM) (*Ashburner and Friston, 2000*; *Ashburner and Friston, 2005*) (http://www.fil.ion.ucl.ac.uk/

spm) pipeline to extract cortical gray matter probability maps as well as maps of cortical thickness (CT) and pial area (PA)(*Dale et al., 1999*; *Fischl et al., 1999*) estimates by means of transforming all anatomical T1-weighted cortical surfaces through FreeSurfer v5.3 (http://surfer.nmr.mgh.harvard. edu). Further, the diffusion-weighted MRI data were used to extract several features, that is fractional anisotropy (FA), anisotropy mode (MO) and mean diffusivity (MD) (*Smith et al., 2006*; *Jenkinson et al., 2012*) (https://fsl.fmrib.ox.ac.uk/fsl/v5.0.9). In addition to these structural readouts and in order to also include purely local morphometric differences across subjects, we also consider the images containing the Jacobian determinants (JD) of the warp fields defining the transformations of each subject's structural image onto a reference brain. These feature extraction operations are schematically summarized in *Figure 1* operation A, and full details on the data processing performed to achieve each feature are provided in the Appendix 1 under the section '*Individual features preprocessing*'. From the initial N = 500 participants, several subjects were excluded on the basis of abnormalities in any of the features. In total, N = 448 subjects were entered into further analyses. We then use the Linked-ICA model (*Groves et al., 2011*) to simultaneously factorize the considered N = 448 subjects' VBM, FA, MO, MD, CT, PA and JD features into independent sources (or components) of spatial variation. In brief, Linked-ICA is an extension of Bayesian ICA (*Choudrey, 2002*) to multiple input sets, where all individual ICA factorizations are linked through a shared common mixing matrix that reflect the subject-wise contribution to each component. This operation is represented in *Figure 1* operation B where we can also appreciate that such factorization provides - per component - a set of spatial maps (one per feature modality), a vector of feature loadings that describe the degree to which the component is 'driven' by the different modalities, and a vector that describes how each individual subject contributes to a given component. Importantly, the subject-loadings define the cross-subject variation of the multi-modal effects and can subsequently be used to study relationships to other behavioral or demographic cross-subject variations by means of simple correlations. All mathematical derivations involved in the Linked ICA factorization can be found at the original paper describing the algorithm (*Groves et al., 2011*). Given our sample size and following (*Groves et al., 2011*; *Groves et al., 2012*) we report full results from a 100 dimensional factorization. Different model order decompositions were also performed to demonstrate the robustness to the choice of dimensionality. Further, we also performed two analogous multi-modal analyses, one including resting state fMRI data and another excluding the JD features, as well as an independent component analyses (*Beckmann and Smith, 2004*) of the JD features in isolation to evaluate the dependency of the results on purely morphometric differences- these are reported in Appendix 1.

Further details and code implementing each feature extraction procedure as well as the Linked ICA factorization are publicly available at *Llera (2019)* (copy archived at https://github.com/elifesciences-publications/Llera_elife_2019_1).

## Statistical analysis

To uncover relationships between the behavioral/demographic measures and the components obtained from the Linked-ICA decomposition we perform a correlation analysis between each independent component subjects' contribution and each available behavioral measure. This operation is schematically summarized in *Figure 1* operation C. To take into account the family structure present in the HCP sample while assessing significance we use the Permutation Analysis of Linear Models (PALM) (*Winkler et al., 2014*; *Winkler et al., 2015*) and use $10^6$ permutations per tested correlation (*Figure 1* operation D). We define significance at p<0.05 and address the multiple comparison by applying FDR correction (*Benjamini and Hochberg, 1995*) as well as full Bonferroni correction (*Figure 1* operation E).

## Acknowledgements

Data were provided [in part] by the Human Connectome Project, WU-Minn Consortium (Principal Investigators: David Van Essen and Kamil Ugurbil; 1U54MH091657) funded by the 16 NIH Institutes and Centers that support the NIH Blueprint for Neuroscience Research; and by the McDonnell Center for Systems Neuroscience at Washington University.

We are grateful to Stephen M Smith for the helpful discussions and for sharing with us their results as presented in *Smith et al. (2015)*. We would also like to thank V Kumar for help with the

visualization of the DWI results and to Paula C Salamone for the help with the graphics. The research leading to these results has received funding through the developing Human Connectome Project (dHCP), a Synergy Grant by the European Research Council under the European Union's Seventh Framework Programme (FP/2007–2013), ERC Grant Agreement no. 319456. We further gratefully acknowledge support from the Netherlands Organization for Scientific Research (NWO) through VIDI grant to CFB (864.12.003) and we also gratefully acknowledge funding from the Wellcome Trust UK Strategic Award (098369/Z/12/Z).

## Additional information

### Funding

| Funder | Grant reference number | Author |
|---|---|---|
| Wellcome Trust | UK Strategic Award 098369/Z/12/Z | Christian F Beckmann |
| Nederlandse Organisatie voor Wetenschappelijk Onderzoek | 864.12.003 | Christian F Beckmann |
| EU Seventh Framework Programme | Synergy Grant ERC Grant Agreement no.319456 | Christian F Beckmann |

The funders had no role in study design, data collection and interpretation, or the decision to submit the work for publication.

### Author contributions

Alberto Llera, Conceptualization, Software, Formal analysis, Validation, Investigation, Visualization, Methodology, Writing—original draft, Writing—review and editing; Thomas Wolfers, Conceptualization, Resources, Software, Formal analysis, Writing—original draft; Peter Mulders, Conceptualization, Validation, Investigation, Writing—original draft; Christian F Beckmann, Conceptualization, Supervision, Funding acquisition

### Author ORCIDs

Alberto Llera (iD) https://orcid.org/0000-0002-8358-8625

### Ethics

Human subjects: HCP data were acquired using protocols approved by the Washington University institutional review board. Informed consent was obtained from subjects. Anonymised data are publicly available from ConnectomeDB (db.humanconnectome.org; Hodge et al., 2016). Certain parts of the dataset used in this study, such as the age of the subjects, are available subject to restricted data usage terms, requiring researchers to ensure that the anonymity of subjects is protected (Van Essen et al., 2013). Informed consent and consent to publish was obtained from the Human Connectome Project according to the declaration of Helsinki. Research conducted at the Donders Center for Cognitive Neuroimage is covered by the protocol approved by the 'Commissie Mensgebonden Onderzoek (CMO) Regio Arnhem-Nijmegen' registered under CMO number 2014/288.

### Decision letter and Author response

Decision letter https://doi.org/10.7554/eLife.44443.024
Author response https://doi.org/10.7554/eLife.44443.025

## Additional files

### Supplementary files

• Supplementary file 1. Summary of the behavioral/demographic measures present in the HCP sample.
DOI: https://doi.org/10.7554/eLife.44443.006

• Supplementary file 2. Significant results. First column presents the component number, second the behavioral or demographic measure it correlates with and third and fourth columns present the correlation value and the permutation p-value. Significance is defined at p<0.05 and we used FDR correction for multiple correction (q < 2.2×10$^{-4}$).
DOI: https://doi.org/10.7554/eLife.44443.007

• Supplementary file 3. Comparison between the positive-negative mode presented in the main text and the multi-modal analyses excluding the JD feature (right column).
DOI: https://doi.org/10.7554/eLife.44443.008

• Supplementary file 4. Summary of the uni-modal analyses using the JD feature. In the second row all relationships are significant after multiple comparison correction. For the uni-modal analysis (the third row), significant associations after multiple comparison correction are denoted with a double asterisk and nominal significant but not significant after multiple comparison correction are marked with a single asterisk.
DOI: https://doi.org/10.7554/eLife.44443.009

• Supplementary file 5. Linked ICA spatial maps associated with the FA feature.
DOI: https://doi.org/10.7554/eLife.44443.010

• Supplementary file 6. Linked ICA spatial maps associated with the MD feature.
DOI: https://doi.org/10.7554/eLife.44443.011

• Supplementary file 7. Linked ICA spatial maps associated with the MO feature.
DOI: https://doi.org/10.7554/eLife.44443.012

• Supplementary file 8. Linked ICA spatial maps associated with the VBM feature.
DOI: https://doi.org/10.7554/eLife.44443.013

• Supplementary file 9. Linked ICA spatial maps associated with the JD feature.
DOI: https://doi.org/10.7554/eLife.44443.014

• Supplementary file 10. Linked ICA spatial maps associated with the left hemisphere CT feature.
DOI: https://doi.org/10.7554/eLife.44443.015

• Supplementary file 11. Linked ICA spatial maps associated with the right hemisphere CT feature.
DOI: https://doi.org/10.7554/eLife.44443.016

• Supplementary file 12. Linked ICA spatial maps associated with the left hemisphere PA feature.
DOI: https://doi.org/10.7554/eLife.44443.017

• Supplementary file 13. Linked ICA spatial maps associated with the right hemisphere PA feature.
DOI: https://doi.org/10.7554/eLife.44443.018

• Transparent reporting form
DOI: https://doi.org/10.7554/eLife.44443.019

## Data availability

All data analysed during this study are anonymised and publicly available from ConnectomeDB (db. humanconnectome.org; Hodge et al., 2016). It can be freely downloaded after creation of an account at "https://db.humanconnectome.org/app/template/Login.vm". Certain parts of the dataset used in this study, such as the age of the subjects, are available subject to restricted data usage terms, requiring researchers to ensure that the anonymity of subjects is protected (Van Essen et al., 2013). Relevant data generated by the analyses we performed are included in the manuscript and supporting files. Further details can be found at https://github.com/allera/Llera_elife_2019_1 (copy archived at https://github.com/elifesciences-publications/Llera_elife_2019_1).

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

# Appendix 1

DOI: https://doi.org/10.7554/eLife.44443.020

In *Supplementary file 1* we provide a summary of the behavioral and demographic measures present in the Human Connectome Project (HCP) sample. For easier interpretation we grouped them here by categories and a full detailed description can be found in van Essen et al. (*Van Essen et al., 2012*).

## Individual features pre-processing:

The structural data was preprocessed using the computational analysis toolbox computational analysis toolbox (CAT)−12 (http://dbm.neuro.uni-jena.de/cat/) (*Nenadic et al., 2015*) which is an extension of Voxel Based Morphometry in statistical parametric mapping (SPM) (*Ashburner and Friston, 2000*). CAT-12 uses an internal interpolation to provide more reliable results than SPM12-VBM. It extends standard SPM processing by for instance including different denoising methods and a modified brain extraction procedure. Prior to gray matter volume estimation, all participants' T1 images were affinely aligned. Subsequently, images were segmented, normalized, and bias-field-corrected (*Elam and Van Essen, 2013*; *Ashburner and Friston, 2000*), yielding images containing gray and white matter segments plus CSF. DARTEL (*Ashburner, 2007*) was then used to normalize all images to a standard gray matter template provided by CAT-12. Subsequently, all gray matter volumes were smoothed with a 9.4 mm FWHM Gaussian smoothing kernel (sigma = 4 mm).

Structural MRI images were processed with the FreeSurfer v5.3 software to extract measures for cortical thickness and areal expansion (*Dale et al., 1999*; *Fischl et al., 1999*) (http://surfer.nmr.mgh.harvard.edu). The standard FreeSurfer preprocessing pipeline (recon-all) was applied to these images, in which a reconstruction of the cortical sheet was estimated using intensity and continuity information. Cortical thickness was determined as the closest distance from the gray/white boundary to the gray/cerebrospinal fluid (CSF) boundary at each vertex (*Fischl and Dale, 2000*). Surface area in FreeSurfer is estimated as relative amount of expansion or compression at each vertex when registering each participant's surface to a common atlas. Surface maps were resampled and mapped to a common coordinate system (*Fischl et al., 2008*). During preprocessing, the data were registered onto the high-resolution average participant surface space (fsaverage), and a 10 mm FWHM surface-based smoothing kernel was applied.

Further, the Jacobian images for each subject are directly available from the HCP repository and the diffusion weighted data (DWI) was preprocessed using the DTIFIT routine from FSL (*Jenkinson et al., 2012*; *Ashburner, 2007*) (https://fsl.fmrib.ox.ac.uk/fsl) to create the FA, MO and MD images that were then feed into the TBSS pipeline (*Smith et al., 2006*).

Finally, for computational reasons (*Groves et al., 2011*; *Groves et al., 2012*), the VBM images were spatially down sampled to 4 mm isotropic and the DWI images to 2 mm isotropic voxels.

To perform structural-functional integration we used partial correlation matrices obtained from resting state fMRI. More concretely we considered a 200-dimensional group ICA decomposition followed by dual regression to extract individual spatial maps and time courses. These data as well as more detailed pre-processing details are publically available at https://db.humanconnectome.org/data/projects/HCP_1200. We then computed regularized partial correlation matrices (size 200 × 200), remove diagonal and redundant elements due to symmetry, and vectorize them for each subject. Finally, we built a functional MRI partial correlation feature matrix by putting each subject partial correlation vector on a column and use this matrix as input for the Linked ICA factorization together with all previously considered structural features.

Further details and code implementing all the operations described here can be found at *Llera (2019)*.

## Image Quality Assessment (QA)

All structural images were checked on the basis of their associated quality measures calculated by CAT-12 toolbox and eyeballed by experts on quality. Diffusion images were rejected from the analyses by expert's visual inspection of the registration, brain extraction and results of the DWI pipeline (*Jenkinson et al., 2012*; *Ashburner, 2007*). Resting state fMRI data was corrected for motion using FSLFIX; for further details on the resting state preprocessing we refer the reader to https://db.humanconnectome.org/data/projects/HCP_1200.

## Behavioral data processing

We used the restricted behavioral data as provided by the HCP consortium. For each subset of subjects considered at each analyses we removed all behavioral measures that were not available for more than a 5% of subjects. For the remaining measures, missing values were inputted by substituting the missing values for the mean all other subjects in the subset. Data was demeaned before further processing.

## Main results

In *Supplementary file 2* we summarize the significant results obtained (FDR corrected $q < 2.2 \times 10^{-4}$). From left to right columns we present the component number, behavioral/demographical measure, correlation value, and the permutation p-value (PALM).

## Feature modalities relative contribution to components

In *Appendix 1—figure 1* we color code the relative contribution of each feature modality to the components that show at least one Bonferroni corrected significant relationship to behavior or demographics measures ($q < 1.4 \times 10^{-6}$).

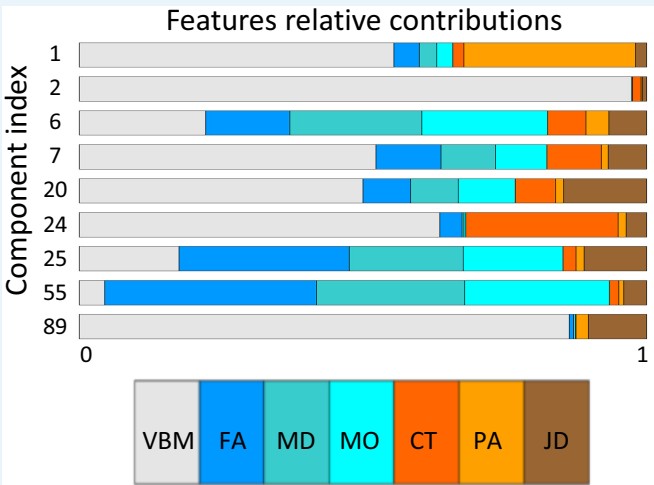

**Appendix 1—figure 1.** Relative contributions of each feature modality to the most relevant components.
DOI: https://doi.org/10.7554/eLife.44443.021

## Robustness: model order

In this section we assess the robustness of the results with respect to the model order choice. To that end we perform a correlation analyses between the reported 100 dimensional factorization subjects-mode and that of a 90 and a 110 dimensional factorizations. In the top row of *Appendix 1—figure 2* we present correlation matrices between a 100 dimensional factorization (y-axis) and a 90 and 110 dimensional factorizations (top left and right panels respectively). Only significant correlations after Bonferroni correction are reported (that is p-value smaller than 0.05 /(100 × 90) and 0.05/ (100 × 110) respectively). For each component of the 100 dimensional factorization we present in the bottom row of *Appendix 1—figure 2*

the absolute correlation value with each of the components of the different dimensionality factorization. We appreciate that most components are recovered with high accuracy (r close to 1) independently of the order of the factorization. The black dashed lines represent the most relevant of the reported components, independent component number 6.

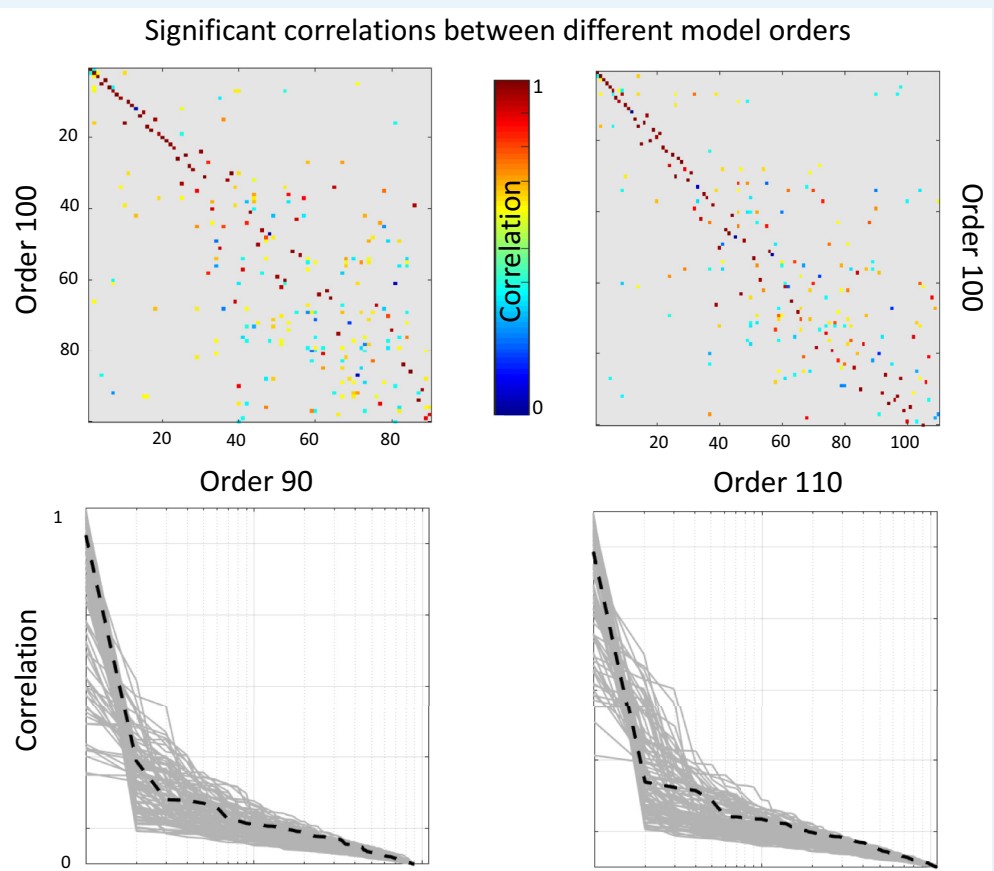

**Appendix 1—figure 2.** Top: Significant correlations between the reported (100 dimensional) factorization and a 90 dimensional (left panel) and 110 dimensional (right panel). Bottom: sorted absolute correlations for each of the components of the reported factorization with the other model orders components. The black discontinuous line represents component number 6.

DOI: https://doi.org/10.7554/eLife.44443.022

## Robustness: analyses without the Jacobians

We present results summarizing the significant findings when performing an analogous analysis to the one reported in the main manuscript without the use of the JD feature. In *Supplementary file 3* we present a comparison between the positive-negative mode as reported in the main text, and the set of behavioral measures significantly associated to component number nine on the new analysis.

Note that component number nine in this analysis (without the JD feature) corresponds to the component number six reported in the main text and it recovers the strongest behavioral associations. Further, this component presented other significant relationships not appearing significant in the main reported mode, for example personality related measures (NEOFAC-C).

## Robustness: Analyzing morphometric differences

In this section we perform an independent component analyses (ICA) factorization (*Beckmann and Smith, 2004*) only of the Jacobian determinant matrices followed by post-hoc correlation analyses with the behavioral and demographic measures. We performed a 100-

dimensional factorization and found a set of 9 components significantly correlating (Bonferroni corrected) with the component number six obtained from the multi-modal Linked ICA analyses. In *Supplementary file 4* we present the correspondence between the positive-negative mode we found through the multi-modal Linked ICA analyses and these nine components. As before, for the multi-modal analyses we only report correlations to behavior significant after FDR correction and for the JD analyses we mark with double asterisk the significant relationships after FDR correction and with a single asterisk the nominal or uncorrected significant relations (p<0.01). We appreciate that although sub FDR significance threshold we observe some correspondence between the purely morphometric differences and the positive-negative mode, these relationships disappear after statistical correction for multiple comparisons.

## On the power of structural and functional associations to behavior

The multi-modal structural analysis we presented here shares 22 behavioral associations with the functional analyses reported in *Smith et al. (2015)*. Paired t-tests, using the absolute correlation values to behavior, revealed no significant difference in r values between the structural and the functional mode at these intersecting behavioral measures. Further, and as expected, the structural mode provides higher correlation values than the functional analyses at the 48 behavioral measures associated to the structural mode (mean r difference = 0.046, p<0.01); on the other side, the functional mode provides higher correlation values than the structural analyses at the 60 behavioral measures associated to the functional mode (mean r difference = 0.048, p<0.01).

To identify the linear dependence between the behavioral/demographic modes obtained from functional and structural data we used a generalized linear model (GLM). We regressed the structural mode from the functional one and performed post-hoc linear correlation analysis of the residualised functional mode relative to behavioral variates as in *Smith et al. (2015)*. Note that structural features – due to the necessary co-alignment within the functional pipelines – acts as a mediator and therefore could induce significant imaging-to-behavior associations (also see *Bijsterbosch et al., 2018*). Conversely, however, the structural features are being analyzed without any possible cross-talk from functional data, so that there is no possible interference from functional to structural features. Post-hoc correlation analysis of the residualised functional mode to behavior revealed a significant decrease in correlation (mean r decrease = 0.078, p<0.01) that result in the structural mode removing 73% of the 60 associations originally found using functional data. The remaining 16 significant relationships involve measures as handedness, education, tobacco use, list sorting, delay discount, or intelligence. As such, the two modes are significantly overlapping.

Although functional data is not required for structural MRI analyses, we also use a GLM to remove the functional effect linearly from the structural mode and also found a significant decrease in r-values (mean r difference = 0.074, p<0.01); in this case remain seven significant behavioral measures that include measures of weight, antisocial behavior (DSM), family structural problems, relational task or adult self-report (ASR) questions.

## Testing structure-function causal effects

We tested the causal effect between the structural mode (component 6) and the functional mode reported in *Smith et al. (2015)* using the model presented in *Hyvarinen and Smith (2013)* and found a functional dependence on structure with a likelihood-ratio measure of 0.0437. This implies a > 20 times higher likelihood of a model of structure-to-function causation relative to the reversed causal model. Note, however, that this only tests these two alternative models relative to each other and does not address the possibility of a third effect (of biological origin) having a causal effect both on brain structure and function (possibly simultaneously). To assess the significance of these findings we use permutation testing. To build a null distribution for this problem we need to break any causal dependence in the data, while keeping similar correlation to the original data (structural and functional modes

correlation). To that end, at each permutation, we interchange the structural and functional mode values for a subset of 210 random subjects while keeping the remaining values fixed; note that 210 is approximately half of the subjects common to both functional and structural analyses. In this way correlation is approximately maintained while breaking any possible causal structure. Then we apply the model presented in *Hyvarinen and Smith (2013)* and obtain a likelihood-ratio measure. We repeated this strategy $10^5$ times to build a null distribution and found that 0.0437 is significant at a p-value=0.0023.

Replication of these causal results was achieved by performing an analogous multi-modal structural Linked ICA analyses considering this time the HCP1200 sample, and including only the subjects not used in the originally considered HCP500 sample. Considering all subjects for whom all structural measures where available, and after QC, this linked ICA factorization was performed using data from 547 subjects. Restricted to the 331 intersecting subjects with the analyses performed in Smith et al., the analyses showed once more a significant correlation between a Linked ICA mode (component 17) and the one reported in Smith et al. (r = 0.1773, p<0.002). Further, consecutive causal analyses from the structural mode to the functional mode estimated a likelihood ratio of ~0.09, that is a significant structure to function causation effect (p<0.005).

