## [Decision Letter]

Thank you for submitting your article "Inter-individual differences in human brain structure and morphometry link to variation in demographics and behavior" for consideration by *eLife*. Your article has been reviewed by three peer reviewers, and the evaluation has been overseen Moritz Helmstaedter as the Reviewing Editor and Richard Ivry as the Senior Editor. The following individuals involved in review of your submission have agreed to reveal their identity: Franco Pestilli (Reviewer #1) and Jason P Lerch (Reviewer #3).

The reviewers have discussed the reviews with one another and the Reviewing Editor has drafted this decision to help you prepare a revised submission.

The reviewers value the importance of the work and see significance in the finding that structural features, not just functional ones, can show high correlation to behavioral phenotypes in humans.

Essential revisions:

1) Additional analyses to solidify the (possibly causal) relation between structural and functional determinants are required as suggested by the reviewers:

a) Including all functional and structural modalities to begin with would at least provide a way of testing their relative contribution to behavioral prediction within the same analytic framework, and/or as suggested (reviewer 2)

b) To strengthen the findings some level of replication from an additional dataset would be helpful (e.g. UK BioBank for older subjects, etc.). (reviewer 3)

c) It would be valuable to try and validate the multivariate imaging components that are seen to predict behavior. For example, are the components of each signature more spatially nested within the canonical brain networks than one would expect by chance? (reviewer 2)

2) Careful revision of the manuscript to enhance clarity and methodological detail (see associated comments by all reviewers below). This especially applies to the figures, which were considered improvable. Please make use of the generous space offering at *eLife*, include more detailed methodological figures where possible (also the supplementary figure can be integrated into the main paper). An illustration of all relevant structural components could help, as well (beyond component number 6).

3) The requests for methodological explanations/clarifications are considered essential for a successful revision. In particular with respect to:

a) Different pre-processing compared to Smith et al., and linked-ICA, reviewer 1);

b) Publication of code on github/gitlab and usability of code (reviewer 1): code and methods need to be available for evaluation, replicability, and reuse;

c) Scale, image QA, motion effects, choice of anatomical features (reviewer 2);

d) Issues of causality and colinearity (reviewer 3, but also raised by the other reviewers).

We are providing the reviewers' full set of comments below for your consideration. However, in line with *eLife’s* consolidated approach, we emphasize that the above requests are the ones we deem essential and ask that you provide a letter detailing your response when you submit the revision. The comments below can be treated as recommendations and a point-by-point response is not required.

*Reviewer #1:*

This is a nice paper following up on previous work by Smith et al., 2015. The results replicate and extend the previously published ones. The manuscript is well written and succinct.

Care should be taken when revising the text as several sections seem unpolished. Yet, I think that several sections of the Materials and methods can improve with added details of the methods. Figure 1 especially should be improved (visually it does not seem to be ready for publication) but also it should be clarified how the dimensionality of the data for each step in the algorithm. It would have to accompany the figure with equations in the Materials and methods describing the mathematical operations and code implementing the model. See below.

A more thorough description of the Linked-ICA is necessary to let this article stand on its own feet. The Materials and methods should provide additional details on the ICA approach, brain and behavioral preprocessing. The detailed description of the preprocessing steps for extractive the behavioral variable should clearly state whether and how the data preprocessing different from Smith et al. If differences existed the authors should clarify why they were necessary and how their choice affected the final result.

In addition, I believe Smith preprocessed/standardized the behavioral and phenotypical data in a way that is not described in the current manuscript. is this correct? Why was the preprocessing performed differently?

Code implementing the analysis should be made readily available. The code should be well documented and allow the readers to reproduce that analyses. The code should help a reader perform the following:

Extract the features as used in the current article starting from each data modality in the HCP release.

Extract and preprocess the behavioral and phenotypical variables given the files that can be obtained by the HCP consortium.

Built the matrices of features as needed for the Linked-ICA modeling.

Perform the additional analysis given the model results.

Reproduce the major plots in the article.

This is especially important in this case as this dataset is public and readers are likely to attempt going beyond the current work. Ideally, the code should be deposited on a platform to allow version tracking and accompanied by a comprehensive readme file describing license and step by step how-to. Github.com seems to be a proper platform for this.

What were the criteria for exclusion for subjects? This is not clearly reported in the Materials and methods.

A few sentences link the following were vague and should be clarified:

"The straight-forward individual linear correlation analysis against the behavioral/demographic measures separately instead affords simple interpretation, albeit possibly being over-conservative given the chosen significance level."

The claim that the current structural mode and Smith functional mode are “strongly correlated” seems an overstatement (r value is only 0.46) I would say that is moderate. Still interesting.

The introductory sentences about phrenology seem out of context unless it is clarified how phrenology fits with the current work.

*Reviewer #2:*

There are several strengths to the work presented. The unbiased modeling of multiple anatomical variables as predictors of multidimensional measures of behavior is valuable, and represents an important complement to the traditional (but likely less biological valid) approach of mass univariate tests of one structural feature against one behavioral measure. A downside of the many-many multivariate analyses is that they can be hard to map back into a lower dimensional space that is easier for us to interpret – but the authors do an excellent job of "translating" the brain-behavior relationships that they find into text and figures that can be more concretely interpreted. The authors also conduct several useful sensitivity analyses which help readers better understand the conditions under which their core findings hold. The authors also quantitatively assess the interrelationship between structural component #6 in their work, and the previously reported multivariate functional imaging component reported in the HCP by Smith et al.

The main potential for novelty and impact of this manuscript (above and beyond the earlier functional imaging study by Smith et al.) rested very heavily on the questions of relative predictive capacity and directional interdependent between multivariate structural (this paper) and functional (Smith et al) predictors of behavior. However, these questions are fundamentally hard to address meaningfully in the absence of longitudinal multimodal data – which would be the ideal observational study design in humans. Appreciate that the authors used components from the Smith paper in causal analyses to conclude in favor of a structure-to-function model (vs. function-to-structure), but I am not confident that this analytic approach can carry the weight being placed upon it. A caveat here however is that I am not qualified to provide an expert statistical review of the Hyvarinen and Smith paper presenting the method used for directional inference. My point is rather a simpler one about limits around the certainty with which one can infer causal processes from cross-sectional data. I wondered if including all functional and structural modalities to begin with would at least provide a way of testing their relative contribution to behavioral prediction within the same analytic framework – even though this would only go some way to getting at the relative predictive utility of structural vs. functional metrics, while still leaving directionality untouched.

I also through the following issues would benefit from further consideration:

The behavioral variables include both raw and age adjusted version for many scales, but age is already a predictor itself. I think it would be good to provide further details around the rationale for selecting which types of scale go in to the multivariate behavior/demographic matrices to be predicted.

It would be good to include more details around image QA and exploration of motion as potential confound.

I appreciate that ratio of variables to observations is an issue, but these analyses would really benefit from a discovery-replication design.

The authors did drop JD to test if structural prediction still there when excluding information about "morphological variation" – but (i) I see this as a specific instance of the more general need to assess relative contribution of different anatomical metrics to different behavioral dimensions., and (ii) opening up the important question of which anatomical features one considers in the first place (for example, no folding or sulfa depth information despite this being provided by FreeSurfer).

It would be valuable to try and validate the multivariate imaging components that are seen to predict behavior. For example, are the components of each signature more spatially nested within the canonical brain networks than one would expect by chance?

*Reviewer #3:*

This is a very interesting article recovering brain-behaviour relations from brain structure in ways that map onto previous findings in brain function. These results are exciting, providing evidence of brain structure influencing (or being influenced by) function. My core reading of the paper leads me to two conclusions:

- I buy that brain structure can predict function – the authors provide solid evidence. To strengthen the findings I would like to see some level of replication from an additional dataset (e.g. UK BioBank for older subjects, etc.).

- I am less convinced of the mediation between structure and function, and especially the claimed link that methodological/misalignment issues might be the cause. That argument could be dropped without weakening the paper; if the authors feel strongly about this point then they need to make a better case.

More detailed comments:

- I don't understand footnotes 1 and 2. The authors chose an FDR threshold of q < 2.2x10e-4? That seems arbitrary? Or does a q < 0.05 correspond to an uncorrected p<2.2x10e4?

- It is curious that VBM would explain much more variation in component 6 than thickness or surface area. It is not easy to determine why from the figure, though part of the explanation could be that subcortical regions play a strong role.

- It is even more curious that JD explains so much less in comp 6 than VBM. What type of VBM was conducted – were the tissue densities modulated by the Jacobians?

- Components 1 and 2 have a strong gender contribution and show significant VBM contributions. How were variations in overall brain volume accounted for, if at all?

- The discussion in the fourth paragraph of the Results confuses me. The authors appear to imply that only the Jacobian determinant measures uniquely morphometric differences; how is it that thickness, surface area, and VBM do not reflect morphometric differences? And aren't these results more of an argument that non-linear registrations in this study were either not tuned very well or that alignment of ideosyncratic cortical features is a hard problem?

- The section on linking the structural and functional analyses is problematic. Given that they correlate using the linear model will obviously run into issues of colinearity. This can be seen by the bivariate nature of the results – covaring structure on function or function on structure gives a similar change in r and removes multiple findings. The secondary argument that the reason why covarying structure removes so many of the function findings – due to misalignment or similar methodological issues – is thus suspect and needs to be expanded on.

- I would like the authors to expand on the advantages of linear models over CCA or other multivariate analyses. In that vein, the Smith et al. paper often referred to in this manuscript also tested structural associations and found them much less relevant than rsfMRI – why the results are different should be discussed.

---

## [Author Response]

Essential revisions:1) Additional analyses to solidify the (possibly causal) relation between structural and functional determinants are required as suggested by the reviewers:a) Including all functional and structural modalities to begin with would at least provide a way of testing their relative contribution to behavioral prediction within the same analytic framework, and/or as suggested (reviewer 2)

The main point of our original submission is to demonstrate that the putative relationship between functional observations and behavioral/demographic variates is already detectable using exclusively structural data modalities. We therefore think that it is important to remain focused on this point by presenting the analysis of structural features without additional functional features first. Nevertheless we agree that the question raised by the reviewers is an important downstream question that naturally results from this primary finding. As suggested we therefore performed an extra analysis using the same Linked ICA methodology, but integrating an additional set of functional features along with the original set of structural ones. In order to fully map on to the Smith et al. paper these functional features are the partial correlation matrices obtained from the two-hundred dimensional group ICA (soft) parcellation performed in resting state fMRI data (publicly available at https://db.humanconnectome.org/data/projects/HCP_1200) and used for the main analysis in Smith et al.

This structural-functional Linked-ICA decomposition recovered the positive-negative mode reported in our original submission; we found a component significantly correlating (r=0.89, p<10^-5) with the mainly reported structural mode (component 6). The contribution of each modality to this mode equals 20% for VBM, 15.6% for FA, 24.4% for MD, 23.9% for MO, 7% for CT, 3% for PA, 5% for JD and 0.0012% for the functional partial correlation feature.

While all structural features provide approximately the same contribution as in the original analyses, it is interesting that the functional data does marginally contribute to this mode, suggesting that structure on its own can explain the positive-negative behavioral mode.

These results have been added to the ninth paragraph of the Results section.

*b) To strengthen the findings some level of replication from an additional dataset would be helpful (e.g. UK BioBank for older subjects, etc.). (reviewer 3)*

We agree that a further validation into another sample would be of high value. Validation of the reported results using the UK BioBank sample is not possible due to the absence of the right battery of behavioral measures, making it impossible to find the functional mode as the one reported in Smith et al., the reason being that such mode is learned using Canonical Correlation Analyses including both functional and behavioral data.

As an alternative to the UK BioBank proposal we considered the full HCP1200 sample. Our previous results were generated from the HCP-500 release of the project. Therefore we computed all structural features necessary and run an Linked ICA analyses analogous to the originally reported, selecting the subjects not used in our original submission. Considering only subjects for which all structural measures where available, and after QC, this additional analysis was performed using data from independent 547 subjects. From these 547 subjects, 331 where common to the analyses performed in Smith et al., and again we identified a single component significantly correlating with the functional positive-negative mode reported in Smith et al. (r = 0.1773, p < 0.002). Causal analyses from this structural Linked ICA mode to the functional estimated a likelihood ratio of ~0.09, i.e. confirming a significant structure to function causation effect (p<0.005, using permutation testing).

These results have been added to the last paragraph of the Results section.

c) It would be valuable to try and validate the multivariate imaging components that are seen to predict behavior. For example, are the components of each signature more spatially nested within the canonical brain networks than one would expect by chance? (reviewer 2)

In our analysis we consider structural multi-modal characterizations of the brain where each modality contains unique information and together builds into a multi-modal multivariate component. As such, these results cannot be nested into the same space as (functional) canonical brain networks e.g. all DWI-derived features will necessarily be focused on white-matter with little to no overlap with canonical gray-matter brain networks. However, the structural weighting in grey matter modalities in orbitofrontal and temporal cortex, in conjunction with the white matter tracts that connect those regions, is a clear indication of an underlying network structure. We have emphasized this information in the revised manuscript (Results, second paragraph) and made available the full NIfTI images for readers to download.

2) Careful revision of the manuscript to enhance clarity and methodological detail (see associated comments by all reviewers below). This especially applies to the figures, which were considered improvable. Please make use of the generous space offering at eLife, include more detailed methodological figures where possible (also the supplementary figure can be integrated into the main paper). An illustration of all relevant structural components could help, as well (beyond component number 6).

We revised the full manuscript to improve clarity and added more methodological details (see also response to the next point below). Further, all images quality have been improved and the main figure of the supplementary material has been included in the revised manuscript as Figure 3. The spatial extent of all involved components has also been integrated into the main text of the revised manuscript (Results, third paragraph). We would further like to point out that the improved version of Figure 1 is a clear summary of a very complex analyses pipeline and results. We have added in the revised manuscript more references to the original methods paper where different graphical representations and all formulae can be found, but we certainly believe that the presented Figure 1 summarizes perfectly the process performed.

3) The requests for methodological explanations/clarifications are considered essential for a successful revision. In particular with respect to:a) Different pre-processing compared to Smith et al., and linked-ICA, reviewer 1);

Indeed, the data pre-processing in our original manuscript is different from Smith et al. Our previous submission is using structural rather than functional MR data. In our revised version we made sure that for the functional data we map our pre-processing to the once used in Smith et al. In the revised manuscript we clarified this. With respect to the different pre-processing of behavioral data with respect to Smith et al. we would like to clarify that Smith et al. needed to standardize the data before running the CCA to ensure that different scaled behavioral variables are not biasing the CCA analyses towards a subset of behavioral readouts. This is not the case in our case since the different behavioral variates enter our analyses separately through calculating simple linear correlations. As such, no harmonization across the different measures is required. We added a section in the SM where we detail how behavioral data were pre-process before entering the linear correlation.

In the revised manuscript we further expand on the description of the Linked-ICA factorization approach and clarify that all code and model derivations are already available as part of Groves et al., 2011. This is included in the revised manuscript (Introduction second paragraph and Materials and methods, first paragraph).

b) Publication of code on github/gitlab and usability of code (reviewer 1): code and methods need to be available for evaluation, replicability, and reuse;

All code has been made publicly available as a github repository (https://github.com/allera/Llera_*eLife*_2019_1) and detailed instructions have been provided. The code includes scripts to extract all features, construct all matrices, smooth the data, perform the Linked ICA factorization, perform post-hoc statistics and reproduce the main figures.

This has also been clearly stated in the revised manuscript (Materials and methods, last paragraph).

c) Scale, image QA, motion effects, choice of anatomical features (reviewer 2);

We have now added a section in the SM where we provide additional details. In summary:

Scale: The Linked ICA model is able to automatically estimate and deal with different scales or spatial degrees of freedom across modalities.

Image QA: The structural data was preprocessed with the computational analysis toolbox (http://dbm.neuro.uni-jena.de/cat/) [Nenadic et al., 2015], which is based on VBM analysis in SPM [Ashburner et al., 2000]. This pipeline generates a number of QC measures which were checked and each individual image was manually inspected for gross artefacts. All images that were of good quality were included in the analyses. All DWI images were also visually inspected for anomalies and only subjects with proper structural and DWI data were included in the Linked-ICA factorization.

Motion effects: Motion artefacts in the structural and DWI measures are addressed by directly removing the affected subjects after automatic and visual QC assessment. The motion effects in the resting state fMRI data were addressed using FSLFIX in precisely the same fashion as in Smith et al.

Choice of anatomical features: All anatomical features, with the exception of the Jacobians feature, were selected based on our extensive experience working with the Linked ICA model. Note that in previous publications [Douaud et al., 2014, Francks et al., 2016, Wolfers et al., 2017] we validated the use of the selected set of features to relate to behavioral measures. The feature set has not been further optimized, the first analyses we performed provided the reported results.

d) Issues of causality and colinearity (reviewer 3, but also raised by the other reviewers).

We agree with the reviewers that causal estimation in general, and on imaging data in particular, is not an easy problem. In fact, the causal analyses we included in the previous manuscript is a consequence of editorial requirements to allow the paper to be sent out for review. We believe we have been very careful in reporting such findings and warning on the interpretation by embedding them into a permutation test for significance assessment and stating clearly that “care is advised when considering causal inference on two vectors of observations, as we cannot exclude the possibility that unobserved underlying processes simultaneously influence brain structure and function (‘hidden causation’).”

With respect to the collinearity issue, we would like to clarify that the model we used for causal assessments (Hyvarinen and Smith, 2013) is based on high order statistics, and, to claim a causal relationship, considers the residuals after linear modelling the pair of signals. Consequently, the collinearity between the two variates is accounted for prior to further causal inference. This is now further clarified in the revised version (Discussion, last paragraph).

We are providing the reviewers' full set of comments below for your consideration. However, in line with eLife’s consolidated approach, we emphasize that the above requests are the ones we deem essential and ask that you provide a letter detailing your response when you submit the revision. The comments below can be treated as recommendations and a point-by-point response is not required.

Reviewer #1:

This is a nice paper following up on previous work by Smith et al., 2015. The results replicate and extend the previously published ones. The manuscript is well written and succinct.Care should be taken when revising the text as several sections seem unpolished. Yet, I think that several sections of the Materials and methods can improve with added details of the methods. Figure 1 especially should be improved (visually it does not seem to be ready for publication) but also it should be clarified how the dimensionality of the data for each step in the algorithm. It would have to accompany the figure with equations in the Materials and methods describing the mathematical operations and code implementing the model. See below.

In the revised manuscript all figures have been improved. We decided to do not include further dimensionality details or mathematical derivations of the Linked-ICA model since these were already reported Groves et al., 2011. We clarified this in the revised manuscript (see 3 a).

A more thorough description of the Linked-ICA is necessary to let this article stand on its own feet. The Materials and methods should provide additional details on the ICA approach, brain and behavioral preprocessing. The detailed description of the preprocessing steps for extractive the behavioral variable should clearly state whether and how the data preprocessing different from Smith et al. If differences existed the authors should clarify why they were necessary and how their choice affected the final result.In addition, I believe Smith preprocessed/standardized the behavioral and phenotypical data in a way that is not described in the current manuscript. is this correct? Why was the preprocessing performed differently?

Pre-processing of the behavioral data only differs in terms of the presence (Smith et al) or absence (here) of a cross-measurement normalization stage. In Smith et al. a CCA approach is used to assess associations, requiring harmonization across scales that natively can vary substantially in terms of their range. In our case we calculate correlations of the linked ICA subject mode vector against the interindividual variations of each behavioral measure separately. No cross-measure is therefore required and we can simplify analysis and interpretation.

Code implementing the analysis should be made readily available. The code should be well documented and allow the readers to reproduce that analyses. The code should help a reader perform the following:Extract the features as used in the current article starting from each data modality in the HCP release.Extract and preprocess the behavioral and phenotypical variables given the files that can be obtained by the HCP consortium.Built the matrices of features as needed for the Linked-ICA modeling.Perform the additional analysis given the model results.Reproduce the major plots in the article.This is especially important in this case as this dataset is public and readers are likely to attempt going beyond the current work. Ideally, the code should be deposited on a platform to allow version tracking and accompanied by a comprehensive readme file describing license and step by step how-to. Github.com seems to be a proper platform for this.

We created a GitHub repository containing code and a web-page with all the required instructions (see 3 b).

What were the criteria for exclusion for subjects? This is not clearly reported in the Materials and methods.

We did add a section to the SM of the revised manuscript where we detail the QC procedure.

A few sentences link the following were vague and should be clarified:"The straight-forward individual linear correlation analysis against the behavioral/demographic measures separately instead affords simple interpretation, albeit possibly being over-conservative given the chosen significance level."

We agree that the phrasing could be confusing and opted by remove the end of the sentence. In the revised manuscript reads “The straight-forward individual linear correlation analysis against the behavioral/demographic measures separately instead affords simple interpretation.”

The claim that the current structural mode and Smith functional mode are “strongly correlated” seems an overstatement (r value is only 0.46) I would say that is moderate. Still interesting.

We have changed strongly by significantly.

The introductory sentences about phrenology seem out of context unless it is clarified how phrenology fits with the current work.

We have removed the implicit mention to the phrenology.

Reviewer #2:

There are several strengths to the work presented. The unbiased modeling of multiple anatomical variables as predictors of multidimensional measures of behavior is valuable, and represents an important complement to the traditional (but likely less biological valid) approach of mass univariate tests of one structural feature against one behavioral measure. A downside of the many-many multivariate analyses is that they can be hard to map back into a lower dimensional space that is easier for us to interpret – but the authors do an excellent job of "translating" the brain-behavior relationships that they find into text and figures that can be more concretely interpreted. The authors also conduct several useful sensitivity analyses which help readers better understand the conditions under which their core findings hold. The authors also quantitatively assess the interrelationship between structural component #6 in their work, and the previously reported multivariate functional imaging component reported in the HCP by Smith et al.The main potential for novelty and impact of this manuscript (above and beyond the earlier functional imaging study by Smith et al.) rested very heavily on the questions of relative predictive capacity and directional interdependent between multivariate structural (this paper) and functional (Smith et al) predictors of behavior. However, these questions are fundamentally hard to address meaningfully in the absence of longitudinal multimodal data – which would be the ideal observational study design in humans. Appreciate that the authors used components from the Smith paper in causal analyses to conclude in favor of a structure-to-function model (vs. function-to-structure), but I am not confident that this analytic approach can carry the weight being placed upon it. A caveat here however is that I am not qualified to provide an expert statistical review of the Hyvarinen and Smith paper presenting the method used for directional inference. My point is rather a simpler one about limits around the certainty with which one can infer causal processes from cross-sectional data. I wondered if including all functional and structural modalities to begin with would at least provide a way of testing their relative contribution to behavioral prediction within the same analytic framework – even though this would only go some way to getting at the relative predictive utility of structural vs. functional metrics, while still leaving directionality untouched.

We thank the reviewer for the positive view of our work and for pointing out some valuable critical points for discussion. The causality and structural-functional points have been answered at 1a.

I also through the following issues would benefit from further consideration:The behavioral variables include both raw and age adjusted version for many scales, but age is already a predictor itself. I think it would be good to provide further details around the rationale for selecting which types of scale go in to the multivariate behavior/demographic matrices to be predicted.

Please refer to answer to point 3 a.

It would be good to include more details around image QA and exploration of motion as potential confound.

More details have been added in the SM of the revised manuscript.

I appreciate that ratio of variables to observations is an issue, but these analyses would really benefit from a discovery-replication design.

Please refer to answer to point 1 b.

The authors did drop JD to test if structural prediction still there when excluding information about "morphological variation" – but (i) I see this as a specific instance of the more general need to assess relative contribution of different anatomical metrics to different behavioral dimensions., and (ii) opening up the important question of which anatomical features one considers in the first place (for example, no folding or sulfa depth information despite this being provided by FreeSurfer).

We refer the reviewer to the answer to point 3 c. As a brief resume, the set of selected features, except for the JD, is based in previous Linked ICA analyses we performed that showed these set of features as relevant to relate to behavioral measures publications [Douaud et al., 2014, Francks et al., 2016, Wolfers et al., 2017]. The inclusion/exclusion of the JD feature was intended to test if the deformation fields, that are used to map fMRI data to a spatial common space (MNI), could explain the positive-negative mode in their own. Further mention that the subset of features used have not been optimized.

It would be valuable to try and validate the multivariate imaging components that are seen to predict behavior. For example, are the components of each signature more spatially nested within the canonical brain networks than one would expect by chance?

We refer the reviewer to the answer to point 3 c.

Reviewer #3:

This is a very interesting article recovering brain-behaviour relations from brain structure in ways that map onto previous findings in brain function. These results are exciting, providing evidence of brain structure influencing (or being influenced by) function. My core reading of the paper leads me to two conclusions:- I buy that brain structure can predict function – the authors provide solid evidence. To strengthen the findings I would like to see some level of replication from an additional dataset (e.g. UK BioBank for older subjects, etc.).

We address this point in 1 b.

- I am less convinced of the mediation between structure and function, and especially the claimed link that methodological/misalignment issues might be the cause. That argument could be dropped without weakening the paper; if the authors feel strongly about this point then they need to make a better case.

In the revised manuscript we have more rephrased that part and have removed direct references to the misalignment argument (Results, eighth paragraph and Discussion, second paragraph).

More detailed comments:- I don't understand footnotes 1 and 2. The authors chose an FDR threshold of q < 2.2x10e-4? That seems arbitrary? Or does a q < 0.05 correspond to an uncorrected p<2.2x10e4?

Yes, we mean that p < 0.05 corresponds to q < 2.2x10e-4.

- It is curious that VBM would explain much more variation in component 6 than thickness or surface area. It is not easy to determine why from the figure, though part of the explanation could be that subcortical regions play a strong role.

We agree and have added this to the manuscript.

- It is even more curious that JD explains so much less in comp 6 than VBM. What type of VBM was conducted – were the tissue densities modulated by the Jacobians?

Yes the VBM is modulated with the Jacobians. Details on the VBM pipeline are improved in the revised manuscript.

- Components 1 and 2 have a strong gender contribution and show significant VBM contributions. How were variations in overall brain volume accounted for, if at all?

All analyses are performed in a common spatial space and we did not further account for brain volume differences.

- The discussion in the fourth paragraph of the Results confuses me. The authors appear to imply that only the Jacobian determinant measures uniquely morphometric differences; how is it that thickness, surface area, and VBM do not reflect morphometric differences?

We identified the JD feature as purely morphometric differences because they directly reflect structural disagreement with respect to a template. All other features reflect measures that do not relate directly to differences, but to each individual structural characteristics.

And aren't these results more of an argument that non-linear registrations in this study were either not tuned very well or that alignment of ideosyncratic cortical features is a hard problem?

We checked the registration and rejected subjects based on that criteria. We further agree that alignment of ideosyncratic cortical features is a hard problem, which is in fact a problem for most structural as well as functional analyses.

- The section on linking the structural and functional analyses is problematic. Given that they correlate using the linear model will obviously run into issues of colinearity. This can be seen by the bivariate nature of the results – covaring structure on function or function on structure gives a similar change in r and removes multiple findings. The secondary argument that the reason why covarying structure removes so many of the function findings – due to misalignment or similar methodological issues – is thus suspect and needs to be expanded on.

We refer the reviewer to 1 b. Further, in the revised manuscript we have removed direct references to the mis-alignment argument (Results, eighth paragraph and Discussion, second paragraph).

- I would like the authors to expand on the advantages of linear models over CCA or other multivariate analyses. In that vein, the Smith et al. paper often referred to in this manuscript also tested structural associations and found them much less relevant than rsfMRI – why the results are different should be discussed.

The reason why we find different results that Smith et al. is that we use a multi-modal integration approach in this sample. We did not find such strong associations with any unimodal structural analyses.